# A Qualitative Study Evaluating the Factors Affecting Families’ Adherence to the First COVID-19 Lockdown in England Using the COM-B Model and TDF

**DOI:** 10.3390/ijerph19127305

**Published:** 2022-06-14

**Authors:** Lisa Woodland, Ava Hodson, Rebecca K. Webster, Richard Amlôt, Louise E. Smith, James Rubin

**Affiliations:** 1Department of Psychological Medicine, King’s College London, NIHR Health Protection Unit in Emergency Preparedness and Response, London SE5 9RJ, UK; louise.e.smith@kcl.ac.uk (L.E.S.); gideon.rubin@kcl.ac.uk (J.R.); 2Department of War Studies, King’s College London, NIHR Health Protection Unit in Emergency Preparedness and Response, London WC2R 2LS, UK; ava.hodson@kcl.ac.uk; 3Department of Psychology, University of Sheffield, Sheffield S10 2LT, UK; r.k.webster@sheffield.ac.uk; 4Behavioural Science and Insights Unit, UK Health Security Agency, Porton Down, NIHR Health Protection Unit in Emergency Preparedness and Response, Salisbury SP4 0JG, UK; richard.amlot@phe.gov.uk

**Keywords:** COVID-19, children, parents, England, adherence, COM-B, theoretical domains framework, non-pharmaceutical intervention

## Abstract

The ability of families to adhere to public health guidance is critical to controlling a pandemic. We conducted qualitative interviews with 30 parents of children aged 18 and under, between 16 and 21 April 2020 when schools in England were closed due to the COVID-19 pandemic. Using the Theoretical Domains Framework, we classified the factors that influenced adherence to seven non-pharmaceutical interventions. We found 40 factors that influenced a family’s ability to adhere. Parents generally indicated they could adhere and reported how their family had changed their behaviour to comply with the guidance. Parents primarily reported they were motivated to adhere out of concern for the health consequences of COVID-19, and because the guidance was delivered by the government. However, we found that reduced access to resources (e.g., technology, transport, and outside space) and social influences that encouraged non-adherent behaviour, decreased adherence. Furthermore, we suggest that families with low psychological and physical ability may face additional challenges to adherence and need to be supported. During future school closures, public health agencies should account for these factors when developing guidance.

## 1. Introduction

During the early stages of the COVID-19 pandemic, a set of stringent non-pharmaceutical interventions (NPIs) were introduced in England to reduce COVID-19 transmission [1,2,3]. Substantial research shows that many NPIs control disease outbreaks [4,5,6], and evidence supports their use for COVID-19 [7,8,9].

An NPI commonly sits in one of three categories: personal NPIs, such as covering your mouth when coughing, staying at home when symptomatic, and washing hands with soap and water; community NPIs, such as physical distancing in workplaces, schools, and at social events; and environmental NPIs, such as cleaning surfaces. Of note, the language of NPIs is changing, and some organisations, such as World Health Organisation (WHO), have started to use ‘Public health and social measures’ (PHSMs) in place of ‘NPI’ [10]. NPI terminology is still in use and was common throughout the pandemic; therefore, we are using NPI.

In England, the UK government initially promoted “Catch it. Bin it. Kill it,” a campaign introduced in 2013, to reduce infections by adopting good hygiene practices [11] and suggested NPIs such as working from home where possible and limiting social activity (i.e., physical distancing interventions) as optional measures to reduce the spread of COVID-19 [12,13]. On 23 March 2020, a national “lockdown” was announced, mandating several measures through law, including closing schools and non-essential shops and requiring members of the public to stay at home (with exceptions for essential shopping, work, and exercise) [14,15]. The public were also strongly encouraged to remain physically distanced when outside (“you have to stay two meters apart,” HM Government [16]) and to continue good hygiene practices [17,18]. During this time, schools were kept open to allow specific children to attend on a voluntary basis, such as children of keyworkers (children whose parents were working in roles critical to the COVID-19 response) and vulnerable children, such as those with a health care plan [19,20].

When implementing an NPI strategy, the social and economic consequences of the NPI must be considered. In relation to COVID-19, closing schools to implement physical distancing is a prime example of an NPI with a high social and economic cost [21]. Uncertainties include the social, health, and educational impacts on children from closing schools [22,23], transmission rates of COVID-19 between children and staff [24], and the health consequences of having COVID-19 [25,26,27,28]. Adherence to such measures also determines how effective an NPI strategy is in reducing infection [29,30]. Modelling studies suggested that for school closures to be effective, children need to reduce their social contacts and not continue to socialise with others outside of school [31]. A study that simulated influenza found 341 symptomatic cases throughout the epidemic when adherence to ‘stay-at-home’ was 90%; however, this increased to 1551 symptomatic cases when adherence was 50%; in this circumstance, the epidemic lasted twice as long [29]. Our review of 19 studies reporting on actual school closures during emergencies found children sometimes continued to socialise outside the home [32]. Children were often looked after by non-household members, continued to engage in various recreational and social activities, including meeting friends, using public transport, and visiting restaurants, and young children attended supermarkets with their parents [32]. In the UK, most of these opportunities to socialise were restricted during the first lockdown. 

One UK study from July 2020 found that children socialising outside the home increased when parents perceived the risk of COVID-19 as low and that the policy of allowing children of keyworkers to attend school reduced parents’ perception of risk [33]. Children socialising with others also increased when a parent worried about their child’s mental health [33]. Another study reported adherence to guidance was less likely for a parent who was unemployed due to COVID-19, living in an apartment, and if there was a keyworker (working in a role critical to the COVID-19 response) in the household [34]. In addition, greater adherence to the guidance was observed for children compared to their parents.

Adopting NPIs often requires people to make changes to their everyday behaviour. Therefore, frameworks of behaviour change are used to categorise factors that might influence adherence. One such framework used commonly in health contexts is the COM-B model [35]. This suggests that for any behaviour change to occur there must be a change in one or more of the following behaviour components: (1) Capability (personal and psychological), (2) Opportunity (social and environmental), and (3) Motivation (reflective and automatic) [36,37]. It is essential that the intervention not only produces behaviour change but that the change in behaviour is adherent and mitigates unintentional non-adherent behaviours [38,39]. The Theoretical Domains Framework (TDF) is often used in combination with COM-B. The TDF allows for the identification of the influences on a person’s capability, opportunity, and motivation and has 15 theoretical domains that interlink with the COM-B components [40,41,42]: (1) Capability—physical skills, knowledge, cognitive and interpersonal skills, memory, attention and decision processes, and behavioural regulation; (2) opportunity—environmental context and resources and social influences; and (3) motivation—professional/social role and identity, beliefs about capabilities, optimism, belief about consequences, intentions, goals, reinforcement, and emotion [43] (Figure 1 [44]). 

Understanding a family’s ability to change behaviour according to different NPIs is critical, particularly during school closures. Authorities need to incorporate this into their NPI strategy when controlling for COVID-19 and future disease outbreaks. In this study, we used the COM-B model and TDF to evaluate the influences impacting families’ ability to adhere to the guidance in England’s first national lockdown implemented to control COVID-19.

## 2. Materials and Methods

### 2.1. Design 

We conducted one-to-one qualitative interviews with parents of children aged 18 and under. 

### 2.2. Participants

To be eligible, participants needed to be at least 18 years of age, living in England, and the primary caregiver to at least one child (18 years and under) who, from 23 March 2020, was not attending childcare, pre-school, or school because of the national lockdown. We excluded children who attended schools that provided accommodation (boarding schools). 

We subcontracted participant recruitment to a specialist qualitative market research service, Angelfish Fieldwork [45]. The study was advertised through the recruitment company’s social media channels and emailed to their existing opt-in online audience. Advertisement lasted from 7 April until 21 April, and 539 people applied. Screening of potential participants took place daily, and 47 potential participants were screened for eligibility via telephone. We used purposive sampling to ensure a diverse sample, selecting participants according to gender, ethnicity, marital status, employment status, income, level of education, living region, keyworker status, the number of children in the household, and children’s age.

In total, 32 participants were invited to interview. On the day of their interview, one participant cancelled, and the recruitment company cancelled one interview because electronic consent had not been received, resulting in a total sample of 30 participants. 

### 2.3. Interview Outline 

The interview guide was designed to explore children’s adherence to multiple NPIs during school closures [32]. Four parents of children who usually attend school/childcare reviewed our initial interview guide. Based on their feedback, we amended questions, clarifying those which appeared to be challenging to answer. 

We asked open-ended questions about the topic area, which did not follow the COM-B and TDF theoretical frameworks. At the start of the interview, participants were informed there were no incorrect answers and were encouraged to discuss their own experiences of COVID-19. First, we asked participants to “tell me a bit about your child(ren),” prompting details about changes in their social activities and interactions since the school closures. Second, we asked questions about the family’s behaviour inside and outside the home and their ability to follow the current COVID-19 guidance. Third, we asked questions about the participants’ view of the current COVID-19 guidance. Following public involvement from parents suggesting some of our questions were too broad, we adjusted some questions to ensure we asked direct questions in each topic area we explored. The full guide is included in the Appendix A.

### 2.4. Procedure 

Participants received information sheets detailing the aims and objectives of the interviews. Informed consent was provided electronically and again verbally during screening. Two female researchers with qualitative interview experience (LW and AH) conducted the interviews via telephone. We interviewed the 30 participants in quick succession before starting any analysis due to concerns that the NPI guidance would change and to ensure we would reach data saturation [46]. Interviews lasted a mean of 75 min (range: 36–98 min) and took place between 16 and 21 April 2020. All interviews were audio-recorded and transcribed verbatim. Participants were reimbursed for their time with a £40 e-gift card. 

### 2.5. Reporting 

We reported data following the standards for reporting qualitative research: a synthesis of recommendations [47] and the consolidated criteria for reporting qualitative research (COREQ) [48]. 

We asked participants about other aspects of the lockdown to gain further insight into a family’s experience of the COVID-19 pandemic. Another topic that we explored was how families managed COVID-19-like symptoms. We asked questions such as “has your child/ren had coronavirus or coronavirus symptoms,” and “would you find it difficult to cope if they did have to self-isolate.” We reported our findings on this topic elsewhere [49]. 

### 2.6. Analysis

Analysis was conducted using Nvivo version 12 software (QSR International, Melbourne, Australia) [50]. We evaluated five NPIs that were advised [18] and two legally enforced NPIs, introduced in England’s first national lockdown [15], shown in Table 1. 

Using a positivist epistemological position to analyse the data, LW first grouped all participant responses referring to the seven NPIs of interest. Responses describing influences likely to change behaviour were coded and assigned “initial” code names. These initial codes were substantially revised three times, reducing from 289, to 211, to 80 codes, by grouping similar code names. LW linked each code name that was relevant to one of the 15 theoretical domains in the TDF (see Figure 1). The theoretical domains with at least one linked code name were then mapped onto the COM-B components. We generated the factors influencing behaviour drawn from the initial code names within each included TDF domain. RKW, RA, and GJR reviewed the coding at each analysis stage. Any disagreements were discussed as a group until all authors agreed. 

## 3. Results

### 3.1. Participants

Thirty participants with a mean age of 39 (range: 24 to 64 years) were included in the study. Participants were mostly female (n = 20), white (n = 20), married or co-habiting (n = 21), working over 30 h a week (n = 20), educated to degree level or above (n = 18), and a non-key worker (n = 25). All participants had at least one child who was not attending school or childcare because of the pandemic, although six children were not in childcare or school before the closures and were the younger sibling of another relevant child (total children n = 70). The children’s ages ranged from 2 weeks to 18 years, with a mean age of eight (mode: 10 years, n = 8). Further demographic information is presented in Table 2. 

### 3.2. Factors Affecting Adherence

“Guidance” is an overarching term used throughout the results to collectively refer to the seven NPIs drawn from the government guidance (see Table 1), unless specified. We found that 40 factors influenced families’ adherence to the guidance (these are formatted in bold in the narrative results and presented in Table 3). All six components of the COM-B model and 10 theoretical domains from the TDF were relevant to the 40 factors that influenced behaviour change. A list of all 40 factors, each with supporting evidence, is included as Appendix A.

### 3.3. COM-B Component: Physical Capability

#### Theoretical Domain: Physical Skills

There appeared to be no physical difficulty in changing behaviour to adhere to the guidance. Parents indicated the family were ***building on* *skills they already had in place***, and reported using those skills “more,” “regularly,” and with vigilance. This was particularly true with handwashing (NPI 4) and cleaning the home (NPI 7). For example, parents reported cleaning the home more than usual and cleaning items that they would not usually (e.g., touchpoints in the home such as doors), but did not find this physically challenging. 

Similarly, parents and children had the physical ability to stay-at-home (NPI 1), not meet others (NPI 2), and maintain physical distance (NPI 3) using skills they use in everyday life: 


*“Yeah, we’re telling them, ‘Move to the side’. Well, I guess it’s no different to how it normally is when you’re out walking because they’re quite, not wild but they run everywhere, and we’re very ordered, aren’t we, as adults and we like to stay on the left and all that kind of stuff, where they just run everywhere all the time. So, quite often I spend a lot of time saying, ‘Just move over there’, or, ‘Let this person past’, or, ‘Step to the side’. So, that’s still very similar.”*
(P17)

### 3.4. COM-B Component: Psychological Capability 

#### 3.4.1. Theoretical Domain: Knowledge 

Knowledge of the guidance was gained mainly through the media, social interaction, and schools. ***Delivering clear guidance*** increased a family’s capability to understand what they needed to do to change behaviour. Parents consistently referred to the “Stay-at-home” message (NPI 1) as “clear” and routinely reported it as the NPI they were most adhering to. Aside from staying at home, most parents reported changing their behaviours to adhere to: not meeting others (NPI 2), maintain physical distance (NPI 3), washing hands regularly (NPI 4), and cleaning the home (NPI 7). Avoiding touching the face (NPI 5) and covering coughs (NPI 6) were less reported as behaviour changes adopted by families. Therefore, families appeared to be well informed of the guidance and their behaviour change resulted in adherent behaviours.

While parents knew they were allowed to leave the home for limited purposes, including exercising outside once a day (NPI 1), many preferred to stay entirely at home:


*“My littlun, he has been good up until now, but I think, you know, it’s starting to show a little bit now and I was going to say to my husband. ‘I think I’m going to have to just take him a good walk or out on his scooter or something like that’. But as I say, I’ve been trying to avoid it really at the moment. So, that’s as much as we’ve been doing really.”*
(P16)

Furthermore, most parents indicated a maximum amount of time they would exercise outside the home, usually ranging between 30 and 60 min. Some parents mentioned restricting the number of household members who could exercise outside the home at one time and where they were allowed to exercise due to their interpretation of the guidance. Although these behaviour changes were adherent, these differences suggest this NPI was less clear. 


*“…there were a lot of people there [park] but it was safe, we weren’t in contact with anybody else, we were able to social distance, but I just felt that the Government guidance was that they didn’t want people to go to the park, so that’s why we started going round the neighbourhood.”*
(P02)

Adherence to an NPI was more likely if it was perceived as effective. For example, participants largely understood how handwashing (NPI 4) would prevent the spread of COVID-19: 


*[Participant describes the “pepper experiment”, which uses pepper to show how soap and water remove germs] “So, that’s the pepper experiment. Please feel free to pass it on! But that’s about them understanding, ‘Well why am I washing my hands?’ ‘Because the soap helps keep the bad germs away.’ So, she knows that, and we did that experiment, so she got to actively do that. And we’ve also said, when she’s been outside, she needs to wash her hands and that’s just for keeping clean.”*
(P24)

Occasionally, perceptions about the high efficacy of certain actions led some participants to query the logic of adhering in the medium to long-term: 


*“So say I stay in the house for fourteen days and my parents stay in the house for fourteen days, surely we should be able to see each other … so I think those are the kind of things that would be like … if I know that I’ve self-isolated for fourteen days, and they know they’ve self-isolated for fourteen days, we should be able to … and I know that … I don’t know.”*
(P01)

However, ***delivering the guidance by a source the parent and child trusts*** could mitigate this effect, particularly among those with low perceived knowledge:


*“Well listen, it’s, as I said to you earlier, it’s all about safety and if that’s what the Government say, they know what they’re doing, they’re professionals in lots and lots of different things. It’s like me going to the doctor with a common cold or whatever, and then her just saying to me, ‘Just stay indoors, go to bed and take hot drinks.’ She’s the doctor, she knows what’s she’s telling you. I ain’t a doctor. I mean she comes to me [***] I know about that—that’s what my job is—but I’m a great believer you listen to people’s experience and education and study and they’ll know about that.”*
(P28)


*[***] text removed to preserve anonymity*


Parents’ knowledge of the guidance was important in how their children changed their behaviour to adhere to the guidance. Parents were often a ***second-hand source of information***, informing and clarifying misinformation for their children. Parents would routinely query their children’s behaviour and were often the child’s only source of information about the guidance. 

#### 3.4.2. Theoretical Domain: Cognitive and Interpersonal Skills

Parents reported having no issues in understanding the guidance and repeating it to their children when needed. Notably, the “stay-at-home” message was emphasised as simple. Parents mentioned children from as young as 3 years understanding they needed to stay-at-home (NPI 1), were unable to meet others (NPI 2), and to maintain physical distance (NPI 3).

Parents reported needing to remind their children to change their behaviour when they did not adhere appropriately, but children also reminded their parents. Therefore, families with the cognitive ability required to ***remind their family members to change behaviours*** are likely to have high levels of adherence: 


*“Been alright so far, but obviously before we go out I’m always reminding her about keeping her distance and stuff. And she’s understanding it now but I think she’s struggling a little bit.”*
(P10)

However, low levels of cognitive ability were sufficient to understand the guidance and make the necessary behaviour changes. In particular, hygiene practices (NPI 4 and 6) appeared to require a low level of cognitive ability to understand, and even young pre-school children reminded their parents to follow those NPIs. Parents commonly reported that their children were taught how to wash their hands and cover coughs in school before the guidance and considered it standard practice: 


*“And then whenever I walk home she tells me, ‘Daddy, we need to wash our hands, need to wash our hands, need to keep coronavirus away.’ And I’m like ‘OK’, so any time we go out now, we come in, she just goes straight, she go straight to the bathroom, and wash the hands and then dry the hands as well. So since the nursery already laid the foundation so it’s easier for me to build on it.”*
(P03)

#### 3.4.3. Theoretical Domain: Behaviour Regulation

Parents used techniques to facilitate the ***behaviour change becoming a* *habit***, to increase adherence. For example, parents would use routine to increase handwashing behaviours, suggesting their children should always wash their hands after activities (e.g., before and after meals and entering the home). In contrast, parents were less likely to consider avoiding touching the face (NPI 5) to become a habit. 

Families reported actively ***changing shopping habits*** in order to facilitate limiting leaving the home (NPI 1), and were now being mindful of food waste, planning meals, using deliveries from local amenities (e.g., milkman, butchers, and corner shops), and shopping for friends and family to achieve this: 


*“I’m trying to be strategic about just buying milk and bread from my local shop and trying to order things in batches, you know, like OK, so we need fruit and veg this week, or we need meat this week. But no, it’s not been easy.”*
(P06)

Furthermore, families were combining shopping trips with outdoor exercise and work journeys to limit the number of times they left the house in a day. In line with this, parents were also actively ***avoiding areas perceived to be highly populated***, and times when places (e.g., parks and shops) were busy, to maintain physical distance from others (NPI 3). 

### 3.5. COM-B Component: Physical Opportunity

#### Theoretical Domain: Environmental Context and Resources

The government restrictions at the time resulting in ***places being closed*** encouraged families to stay-at-home (NPI 1), increasing adherence. The ***visual changes in the environment***—for example, streets feeling like a “ghost town” or like “another planet”, were a physical indicator of the pandemic, emphasising the unusual environment and reducing the time families spent outside (NPI 1), facilitating adherent behaviour change. The signs in shops, parks, and the street showing the guidance were reminders to families about the behaviours they should be doing, aiding appropriate behaviour change. Parents reported that the ***“nice weather”*** increased their activity outside (NPI 1) and therefore also reduced the ability to physically distance (NPI 3), although activity outside was still lower and ability to physical distance was increased more than if places were open. 

Whilst staying at home reduced the likelihood of meeting other people, ***meeting people in the street*** was still common and led to face-to-face interactions, although these chance meetings were often reported as physically distanced (NP3). The perception that the environment made physical distancing easier resulted in non-adherence, where some would more frequently leave the home (NPI 1): 


*“Yes, I must admit, we often go out more than once a day, only because, as I say, it’s very rural… we’re able to do that because you actually can walk and not see a soul.”*
(P29)

Parents commonly connected their ***home location*** with their ability to maintain physical distance from others (NPI 3), where rural areas were associated with fewer people and wide-open spaces. In contrast, those living in cities and suburbs did not have parks locally and often reported walking the streets for their daily exercise. Because of the limited options available to exercise, the streets appeared busy with locals also exercising. ***Having a* *garden*** was considered a huge resource to families, and parents often reported utilising it to reduce leaving the home (NPI 1), regardless of the either rural or urban local area. Fences and walls within the home were a ***physical barrier*** to mitigate close contact with people outside the home (NPI 3). When physical barriers were not accessible, parents would set boundaries using objects in the environment: 


*“I think literally just saying, ‘You have to stand here’ and if we’re on a path or something, ‘Don’t cross the path’. He’s quite good really—if you set him a boundary, he won’t really pass that.”*
(P09)

Parents also requested their children to keep their hands in their pockets to prevent touching items potentially contaminated with COVID-19. As a result of this behaviour change, an increase in adherence to avoiding touching the face (NPI 5) would likely occur. 

It was apparent that ***technology*** facilitated adherence. Families increased their activity on social media, gaming, and phone to regularly contact friends and family, mitigating wanting to meet others in person (NPI 2). Access to online exercise activities, online religious services, contacting friends, neighbours, and family members to organise shopping for each other and home shopping deliveries reduced the number of times individuals left the home (NPI 1). 

In contrast, families with **low financial resources** reported increased reasons for leaving home. This included leaving the house to collect food packages and school food vouchers, and print schoolwork. Parents also reported that other people (e.g., friends or family) made more shopping trips to buy shopping for parents who could not afford their shopping:


*“The school have been putting together a food parcel every week, so we have to go down to the school to pick that up and then bring that home.”*
(P26)

One parent reported not changing their behaviour to avoid others (NPI 2) because they had no option but to share a car with someone outside the household so they could get to work. This was due to not owning their own car nor having access to alternative transport. 

Nearly all parents regularly bought cleaning products (NPI 7), including hand sanitiser and hand wash (NPI 4). 

A ***lack of* *childcare*** within the home also posed problems for some. It was infrequent, but parents reported using friends, family, and neighbours for childcare while they were food shopping and exercising. In addition, children who were too young to be at home alone had increased activity outside the home (NPI 1), shopping with their parents or waiting outside the shops. 

***Organisations adapting***, such as shops restricting the number of people allowed at one time and creating a one-way system in the shop, facilitated parents’ ability to change behaviours to adhere to maintaining physical distance (NPI 3). When these were not in place, or there was mixed messaging in shops, individuals found this more difficult. 


*“ … they had about two or three workers all stood together stocking up, not doing the two metres apart, and the aisles are really small in that shop, so everyone’s just next to each other and you can’t do the social distancing in the shops, which is really hard. Even if you’re trying to…”*
(P09)

As well as this, some shops restricted the number of items bought per customer; some larger families were unable to purchase all their essential items in one shop and meant they made more shopping trips (NP1). 

The environment offered by workplaces caused similar issues. Parents had limited control over the ability to adhere to maintaining physical distance (NPI 3); some workplaces asked employees to use self-contained rooms and separated tables, although others had no or inadequate measures. However, most parents reported they could work from home (NPI 1) or change their hours to use more convenient transport modes, helping them distance themselves from others (NPI 3). 

### 3.6. COM-B Component: Social Opportunity

#### Theoretical Domain: Social Influences

Influences from friends, family, and societal norms were an important factor in how a family changed its behaviours. This was particularly true for adhering to maintaining physical distance (NPI 3), which goes against general ***social norms*** and instinct. Parents reported feeling “rude” and were “apprehensive of people” and their reactions when maintaining physical distance. Culturally it would be appropriate to step aside to let someone pass or to avoid colliding. However, to actively cross the street or pass, leaving a broad, 2-m gap, could be considered anti-social: 


*“And everybody’s trying to avoid everybody else but also try and be polite, and it’s odd.”*
(P24)

Some parents witnessed altercations between others about appropriate physical distancing highlighting the difficulty. 

In the same way, receiving ***social approval from others*** reassured parents they were behaving appropriately and continued to do so because of this; examples of approval included a smile or head nod from the others. However, a visual approval gesture was not always required; seeing someone else’s behaviour can also provide social approval for a behaviour. For example, one parent cleaned their door with disinfectant and felt reassured to continue this behaviour after witnessing a neighbour cleaning their front door in the same manner.

Furthermore, parents appeared to change their behaviours to those they perceived others would consider as socially desirable over behaviours they felt were appropriate to the NPI. For example, one parent felt unable to take their children shopping because they felt they would “probably be questioned”, although they perceived taking their child shopping as adherent. Another parent rewashed their hands after someone else entered the bathroom because the other person would think they had “not done it properly.” Parents also seemed aware of the potential influence from others, worrying that their children seeing others spending prolonged periods outside made their children question why they should limit their time outdoors (NPI 1). 

Parents also received social approval through verbal communications, particularly from others they already knew. Parents would discuss with friends and family appropriate behaviour changes relating to adhering to the guidance in everyday conversations, which resulted in ***group conformity of behaviours***. Family and friends could influence behaviour change from adherent to non-adherent and vice versa. However, conflict and disagreements still occurred despite apparent social influences, and in these circumstances behaviour change did not occur. Common disagreements were related to when it was appropriate to leave the home (NPI 1) and deciding which items were “essential” to leave home to buy. 

In principle, behaviour predominately changed because of ***authority relations***. When we asked the reason for changing behaviour, parents often stated because they “were told to,” and this was especially true when told to by the government. The perceived power of the authority figure needed to reflect the perceived level of behaviour required for change, i.e., the larger the change, the higher the authority. Parents would cite the government when requesting their children to change behaviour and actively decided their children should watch the guidance announced to increase their ability to influence behaviour change in their children: 


*“ … they’re both aware who Boris Johnson is, that’s the prime minister, that’s the man that runs our country, and he is saying you need to stay at home … he’s an authority figure in their eyes” and he’s the one that’s said this is what you can and can’t do, and they’re complying because it’s not me that’s told them! <Laughs>”*
(P26)

Parents reflected on ***work–power relations***, and the feeling of being unable to disagree with work recommendations, which impeded their ability to adhere such as attendance in work (NPI 1) and safety measures at work were not always adequate for them to adhere to maintaining physical distance (NPI 3). Parents expressed feelings of “relief” when workplaces put acceptable adjustments in place without requesting them. 

Families needing to continue to celebrate ***significant life events*** such as birthdays, funerals, and religious holidays increased the likelihood of non-adherence to the NPIs to stay-at-home, not meet others, and maintain physical distance (NPI 1, 2, and 3). The cancelling of in-person events or moving them online was common. However, parents still reported meeting others to share gifts, and leaving the home to buy items associated with the event (e.g., birthday cake) rather than attending a celebration—except for a funeral, which “was meant to be limited to ten people, but you can’t really stop people coming to an open cemetery.”

Overall, having a ***social network nearby*** increased the chance of non-adherence to NPI 1, 2, and 3. Friends and family would commonly “knock-on” casually when shopping and exercising and call on each other when they needed social support.

Some parents and children were ***searching for social interaction*** and craving in-person interaction as a result of restrictions. As such, participants offered examples of talking to strangers whilst out and about, and their neighbours during the “Clap for Carers” event (a round of applause for key workers that occurred on the doorstep, once a week), risking non-adherence to NPI 1, 2, and 3. ***Volunteering*** and ***shopping for family, friends, and neighbours*** also increased these opportunities for in-person interaction. However, when carrying out these behaviours, parents commonly described trying to keep 2 meters apart. 

### 3.7. COM-B Component: Reflective Motivation 

#### 3.7.1. Theoretical Domain: Beliefs about Capabilities 

Largely, parents preferred to stay at home (NPI 1) because it was an easy instruction to follow: this related to the ***lack of* *control*** over other people’s actions. This in turn increased their belief in their own ***self-efficacy*** (the ability to change their own behaviour) and ***self-confidence*** (doing everything they can to adhere) in them having complete control to stay-at-home guidance:


*“… it’s the thing that I can control the easiest… you can’t guarantee that people are gonna stay two metres away from you. You can’t guarantee that anyone else is gonna adhere to the guidelines. But these are the things I can control.”*
(P13)

#### 3.7.2. Theoretical Domain: Optimism

Parents would commonly reflect optimism in their ***family’s circumstances,*** often describing themselves as “lucky” to have the resources needed to cope. Typically, two mantras emerged. First, parents suggested ***everyone needs to work together*** and change their behaviours, such as they avoided using resources that other people may need (e.g., shopping delivery slots). Second, a ***shared goal,*** such as lockdown lifting and reducing the spread of COVID-19, enhanced families’ motivation to maintain behaviour change: 


*“… this is something that we all need to work together. If I don’t self-isolate myself and I don’t support the government and I don’t support the NHS, and I don’t support the guidelines, then it won’t happen.”*
(P14)

#### 3.7.3. Theoretical Domain: Beliefs about Consequences 

Beliefs about the consequences of not changing behaviour related primarily to **health consequences** (i.e., the risk of COVID-19 and becoming seriously ill) and were consistently reported as reasons for behaviour change. As such, the ***uncertainty of the health implications*** of not changing behaviour motivated individuals to adhere. However, these beliefs about health consequences were not static, and ***perceptions changed over* *time***, reflecting perceptions about the length of time the guidance was in place and the severity of the health effects increased: 


*“When this first kicked off, I was quite lenient … But as time gets on, it’s been going on, I’ve said, I sat him down and said, ‘Look, we really need to be aware of this and we really need to be serious about this, and look what’s happening.”*
(P28)

### 3.8. COM-B Component: Automatic Motivation

#### Theoretical Domain: Reinforcement 

Of the seven NPIs evaluated, two (NPI 1 and 2) were ***legally enforced*** and parents appeared motivated to change behaviours as with other laws. However, this was only reported sporadically as a reason for behaviour change. Furthermore, parents did not appear to separate the legally enforced NPIs from those that were not when they reported their reasons for changing their behaviour. Children were described as following the guidance due to ***parental discipline***:


*“In my opinion it has to be done. So therefore, I’m going to drill that into the children.”*
(P05)

## 4. Discussion

Individuals changing their behaviour to reduce virus transmission is key to controlling a pandemic. Influencing children to change their behaviour can be difficult, particularly in households with multiple children with different abilities. Therefore, it can be challenging for families to adhere to behavioural recommendations. We used the COM-B model and Theoretical Domains Framework (TDF) to classify factors that influenced adherence among families to seven NPIs used to reduce the spread of COVID-19. We identified 40 factors that influenced a family to change their behaviour and these will be discussed against the main elements of the COM-B model (capability, opportunity, and motivation).

### 4.1. Capability

Families appeared capable of adhering to the guidance when they knew the guidance, and found what was required of them clear and simple to follow. In addition, families had the physical skills required to adhere. Indeed, parents reported that even young children could adhere, indicating that the guidance required low cognitive, personal, and physical ability to adhere. 

Whilst one Canadian study suggested that children had greater adherence to stay-at-home guidance than their parents, we did not observe this and found that families appeared to adopt the same behaviour changes within the household [34]. We found families were actively changing their routines to adopt behaviours that they perceived would most improve their family’s adherence to guidance; this included changing shopping habits to ensure they were visiting shops as “infrequently as possible” and were implementing new habits to facilitate fewer shopping trips (e.g., planning meals to reduce food waste). A change in shopping habits has been observed in other countries because of COVID-19 guidance [51,52]. Encouraging routines and new habits of adherent behaviours appears to increase a family’s adherence and could be actively encouraged in future lockdowns.

However, the guidance also resulted in behaviour changes that worried parents. Some parents reported actively encouraging their children to leave home, where permitted within the rules, out of concern for them. These parents reported their children were not leaving home enough, or at all, because they were too worried or anxious about COVID-19. A systematic review of psychosocial consequences of COVID-19 suggested that isolation at home in children can increase anxiety symptoms and individuals with anxiety disorders tended to be preoccupied with excessive handwashing [53]. Similarly, some parents justified their behaviour changes by clarifying that the changes they had made were not to an extreme or obsessive level. We suggest parents who justified a behaviour in this way were concerned that the behaviour was, or would be, perceived as inappropriate. Guidance encouraging people not to “over-adhere” may be important in future lockdowns. 

### 4.2. Opportunity

Issues around opportunity largely influenced the social distancing NPIs (staying at home, not meeting anyone outside the household, and maintaining physical distance from others) that we classified. Unsurprisingly, removing the physical and social opportunities for non-adherence (e.g., non-essential shop closures) increased their family’s adherence. Family adherence to the stay-at-home NPI reduced the opportunity to engage in non-adherent behaviours generally (e.g., they were less likely to meet others in the street, and children exercising in the garden could maintain physical distance from others because of fencing around the home). In contrast, spending more time outside the home increased the opportunity for non-adherence.

Considering physical opportunity, factors that were connected to reduced resources (e.g., lack of childcare and low financial resources) commonly made it difficult for families to adhere. Previous research suggested that young children went grocery shopping with their parents when schools were closed [32]. We also found this to be true, but generally only for parents without available childcare in the home (e.g., single parents and young children with no older siblings). In contrast, some parents reported an increase in childcare resources because of schools being closed and working from home; partners who may have been at work or older siblings that would usually be at school were available to look after younger children. When children did leave the home, we found that children mainly only left home for exercise and no more than once a day, suggesting that they were adherent to stay-at-home guidance. “Adherence” to the stay-at-home NPI we classified was difficult due to the numerous caveats allowing families to spend time outside (e.g., exercise and essential shopping), while the “work from home” requirement for many people increased parental ability to supervise their children and facilitate adherence. 

Results indicate that parents who had a lower household income left home more often for the essential reasons of collecting food vouchers or schoolwork resources, compared to families with more financial resources. Families on lower incomes are also more likely to walk to the shops, suggesting they may need to make more shopping trips than someone with access to transport because they are limited in the number of items they can carry on each shopping trip [54]. In addition, families on low incomes had more difficulty maintaining physical distance from others when outside. Parents who reported living in urban areas perceived local areas as highly populated and walked the streets for exercise because of a lack of local parks. Research suggests children in low-income families are more likely to live in urban rather than rural areas, which have a higher population density and fewer green spaces [55,56,57]. A study in the US suggested that income inequality was associated with more cases and deaths due to COVID-19 [58]. The imbalance we found between family resources could partly explain the increased mortality seen in deprived areas of England due to COVID-19 [59]. 

For families from higher income households, an increase in resources facilitated adherence. First, access to a garden was consistently mentioned as a resource families used to stay at home and minimised the need to leave home for exercise. Furthermore, families reported additional benefits from having a garden as they could adhere to the guidance and socialise with their neighbours. Access to green spaces has been suggested to support mental health during the COVID-19 pandemic [60]. 

Second, access to technology allowed parents and children to socialise yet maintain physical distance from others. Previous research found that parental worries about their child’s mental health increased the likelihood of meeting people from other households [33]. However, this was not observed and the increased access to technology could explain this difference.

Social opportunity influenced a family’s adherence in several different NPIs. This corresponds with existing findings; a review of psychological factors underlying adherence to COVID-19 regulations suggested that individuals can become key actors and leaders in increasing adherence to COVID-19 guidance, by promoting adherent behaviour within their close social circle (e.g., friends and family) [61]. We found that parents regularly discussed how guidance was being adhered to within their social circle. As a result, group norms were formed, which they and their household followed. However, these norms could reflect adherent or non-adherent behaviour. In addition to this, we also found that strangers could influence families to continue adherent behaviours. 

The use of authority and power figures to promote adherence to NPIs was found to be effective for families. Parents commonly referred to “the Government,” “The Prime Minister,” and “Boris Johnson” as reasons for why their family were adhering. However, we should highlight that we observed a difference between family’s reasons for adhering because the government told them to and because they trusted the experts (referring to public health officials). In support of this difference, a UK survey from April 2020 found only 42% of the public trusted the information provided by the UK Prime Minister, compared to nearly double (84–89%) for subject experts or practitioners [62]. Therefore, we suggest that behaviour change due to the government does not necessarily require trust, although trust does appear to increase adherence. 

### 4.3. Motivation

A family’s belief about the health consequences of non-adherence was an apparent influence on their behaviour change. Our study explored adherence to guidance relating to a prolonged and widespread lockdown, simultaneously affecting many parts of society [63]. In addition, the government and media regularly reported the COVID-19 cases and death rates throughout the pandemic, which likely reinforced the negative health consequences of catching COVID-19 [64,65]. These factors may explain the strong influence of beliefs about health consequences that we observed in parents. 

Similarly, we found that adherence tended to increase over time; parents suggested that the longer the pandemic lasted, the more serious the situation. However, our study was conducted in April 2020, relatively soon after the first lockdown was announced, and therefore we cannot suggest how this association may have been affected by continuing restrictions.

A previous study suggested that schools staying open for vulnerable children reduced parents’ perception of their susceptibility to COVID-19, leading to non-adherence [33]. However, we did not observe this relationship; parents did not report that schools being open to some children had influenced their behaviour. We found that the guidance parents perceived to be effective increased their adherence. In support of our findings, a UK study found those who perceived NPIs to be effective were more likely to adhere to social distancing and hand hygiene NPIs for COVID-19 [66]. Furthermore, a US study found people who had a higher perceived threat of COVID-19 infection were significantly more likely to perceive NPIs as effective and had a high commitment to altruism, supporting the factors we found relating to optimism increasing adherence [67]. Research is growing in this area, and further support for the importance of self-efficacy in adherence to COVID-19 guidance has been shown in numerous studies [68,69,70,71]. Jørgensen, Bor, and Petersen [72] conducted a study across eight countries, including the UK, and found self-efficacy was more motivational than threat appraisal. It also found that governmental trust had surprisingly little motivational power, which supports the finding that trust was different from authority in motivating family’s adherence. 

At the time of our interviews, the government had introduced legal consequences (fixed penalty fines) for non-adherence to specific COVID-19 guidance. Between the start of the lockdown and shortly after the interview period (March to May 2020), over 15,000 fines were issued across England [73]. Families rarely reported the law as a motivating factor to adhere. We assume parents felt it was unlikely that police would enforce penalties for non-adherence. Parents often reported that the guidance could be stricter and supported other countries with tighter restrictions, including arresting people for non-adherence. Fines were increased in August 2020 although media coverage reflecting the system’s inadequacy (e.g., the low number of fines that had been paid and police were inconsistent in enforcing the rules) was unlikely to improve parents’ motivation [74,75,76]. The infrastructure needs to be viable for this strategy to be effective. Promoting voluntary adherence via altruistic motivations rather than force may be a more productive route [77,78].

## 5. Limitations

First, the possibility of selection bias may limit our findings. Parents who opted to participate in the study may be particularly motivated to participate in studies about COVID-19. These parents could be more informed of COVID-19 guidance and more adherent to the guidance. Second, we interviewed more parents who were educated to degree level and above (n = 18, 60%), were in some form of work (n = 25, 83% in full-time or part-time), and white (n = 20, 67%). Therefore, further research is needed in minority groups. Third, as well as children of key workers, children with educational, health, and social needs (i.e., children with a health care plan) were permitted to attend school throughout the pandemic. We only interviewed parents of children who were not attending school because of COVID-19, and parents views may differ for parents whose child attended school during this period. Fourth, masks were not mandatory or advised during the interview period and evaluating an NPI about mask-wearing may alter the results. 

## 6. Implications 

Families appeared to have the ability to adhere to the guidance implemented during the UK’s first lockdown. A combination of personal NPIs known to families and additional restrictive measures appeared to be useful to ensure that families could adhere to the guidance. However, it is unlikely that adherence to social distancing and hygiene measures would be as high if community NPIs were not in place, including closing non-essential shops and requiring parents to work from home. Social influences were prominent, and critical authority figures must deliver the information and consistently adhere to ensure public adherence. Parents encourage their child’s adherence, and authorities need to support children who lack parents or close social influences encouraging adherent behaviour. Furthermore, families with less financial and environmental resources require assistance to prevent health disparities between poorer and wealthier groups. 

## 7. Conclusions

This study furthers research into factors affecting adherence to guidance aimed at controlling a disease outbreak. Families may already be adhering to guidance in everyday life. Where adhering to guidance necessitates a change in behaviour, a family’s ability to adhere requires a combination of factors; at a minimum: (i) capability, (ii) opportunity, or (iii) motivation to adopt the behaviour. Policymakers can use the factors we found to improve adherence to NPIs, such as ensuring guidance requires low physical and cognitive ability to understand and encourage optimism and confidence in the capabilities of the individual. When implementing guidance, consideration is needed about influences that may increase adherent behaviour, particularly social and environmental factors, including families in low-income brackets.

## Figures and Tables

**Figure 1 ijerph-19-07305-f001:**
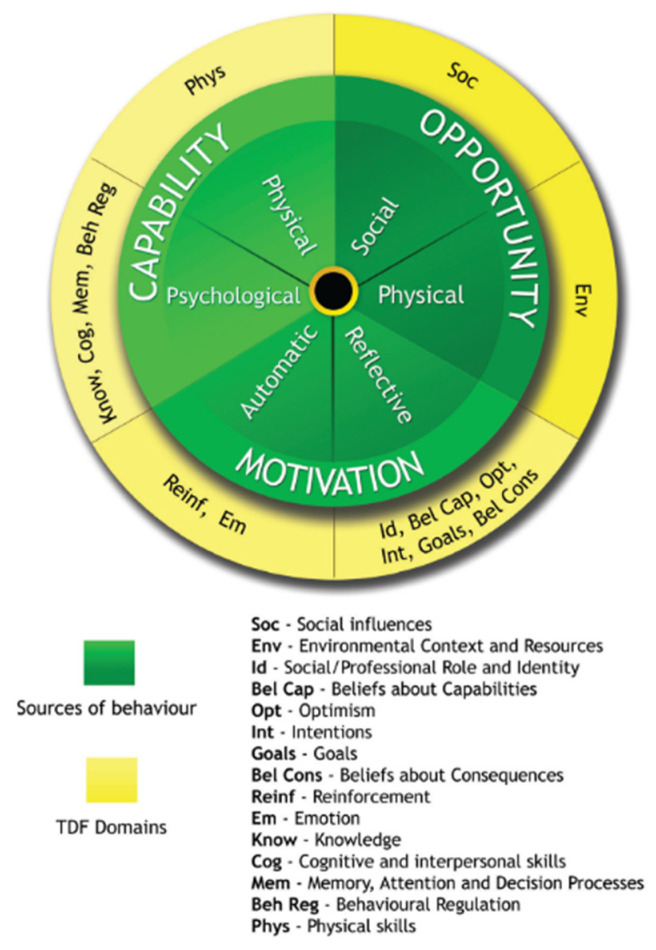
Map of Theoretical Domains Framework (TDF) to source behaviour on COM-B model [44].

**Table 1 ijerph-19-07305-t001:** Non-pharmaceutical interventions (NPIs) evaluated.

NPI Reference Number	Descriptor Used in the Results	NPI Description in the Government Guidance
1 *	Stay-at-home or leave the home for limited purposes	Stay at home You should only leave the home for limited purposes: shopping for basic necessities (e.g., food and medicine), as infrequently as possible one form of exercise a day (e.g., a run, walk, or cycle)—alone or with members of your household any medical need, to provide care or to help a vulnerable person travelling to and from work, but only where this is absolutely necessary and cannot be done from home
2 *	Meeting others	You should not meet friends and family members that do not live in your home
3 **	Physical distance	You should try to stay at least 2 m (3 steps) away from anyone you do not live with
4 **	Handwashing	Washing hands more often—with soap and water for at least 20 s or use hand sanitiser when soap and water is not available
5 **	Avoid touching the face	Avoid touching your eyes, nose, and mouth with unwashed hands
6 **	Covering a cough	Cover your cough or sneeze with a tissue, then throw the tissue in a bin and wash your hands
7 **	Cleaning the home	Clean and disinfect frequently touched objects and surfaces in the home

* Legally enforced in England’s first national lockdown. ** Advised to the public in England’s first national lockdown.

**Table 2 ijerph-19-07305-t002:** Participant (n = 30) and children (n = 70) demographic information.

Demographic Information		Frequency
Gender of Participant	Female	20
	Male	10
Ethnicity of Participant	White	20
	Black, African, Caribbean, or Black British	5
	Mixed or Multiple ethnic groups	3
	Asian or Asian British	2
Marital Status of Participant	Married/Cohabiting	21
	Single/Separated	9
Work Status of Participant	Full-time (Working over 30 h a week)	20
	Part-time (working 8–29 h a week)	5
	Home-maker	3
	Student	1
	Maternity Leave	1
Income of Participant	Under £30,000	12
	£30,000–£50,000	8
	Over £50,000	10
Level of Education of Participant	≤A-level or equivalent	12
	≥Degree or equivalent	18
Living Region of Participant	Yorkshire and the Humber	5
	East of England	4
	Greater London	4
	South West	4
	West Midlands	4
	North West	3
	South East	3
	East Midlands	2
	North East	1
Participant is a keyworker	No	25
	Yes	5
Number of children in the Household	1	9
	2	12
	3	3
	4 and over	6
Age of children (years)	0–4	23
	5–8	13
	9–12	17
	13–16	14
	17–18	3
Usual education setting of children	No childcare	6
	Nursery	6
	Pre-school	5
	Primary	33
	Secondary	16
	Sixth form/College	4

**Table 3 ijerph-19-07305-t003:** Factors (n = 40) that influenced a family’s ability to adhere to seven non-pharmaceutical interventions (NPIs *) implemented in England to control COVID-19 (✕ Adherence to the NPI was not influenced by the domain, ✓ Adherence to the NPI was influenced by the domain).

COM-B Component		Corresponding Theoretical Domain	Factors That Influenced Adherence to the Lockdown Guidance (NPI 1–7, See Table 1)							
				1	2	3	4	5	6	7
Capability	Physical	Physical Skills	Building on skills families already had in place	✓	✓	✓	✓	✓	✓	✓
	Psychological	Knowledge	Delivering clear guidance	✓	✓	✓	✓	✓	✓	✓
			Delivering the guidance by a source the parent and child trusts	✓	✓	✓	✓	✓	✓	✓
			Parents are a second-hand source of information	✓	✓	✓	✓	✓	✓	✓
		Cognitive and interpersonal skills	Reminding family members to change their behaviours	✓	✓	✓	✓	✓	✓	✓
		Behaviour regulation	Behaviour change becoming a habit	✓	✓	✓	✓	✓	✓	✓
			Changing shopping habits	✓	✕	✕	✕	✕	✕	✕
			Avoiding areas perceived to be highly populated	✕	✕	✓	✕	✕	✕	✕
Opportunity	Physical	Environmental context and resources	Places being closed	✓	✕	✓	✕	✕	✕	✕
			Visual changes in the environment	✓	✓	✓	✓	✓	✓	✓
			Nice weather	✓	✕	✓	✕	✕	✕	✕
			Meeting people in the street	✕	✓	✓	✕	✕	✕	✕
			Home location	✓	✕	✓	✕	✕	✕	✕
			Having a garden	✓	✕	✕	✕	✕	✕	✕
			Physical barriers	✕	✕	✓	✕	✓	✕	✕
			Technology	✓	✓	✕	✕	✕	✕	✕
			Low financial resources	✓	✓	✕	✕	✕	✕	✕
			A lack of childcare	✓	✓	✕	✕	✕	✕	✕
			Organisations adapting	✓	✕	✓	✓	✕	✕	✓ **
	Social	Social influences	Social norms	✓	✓	✓	✓	✓	✓	✓
			Social approval from others	✓	✓	✓	✓	✕	✓	✓
			Group conformity of behaviours	✓	✓	✓	✓	✓	✓	✓
			Authority relations	✓	✓	✓	✓	✓	✓	✓
			Work power-relations	✓	✕	✓	✓	✕	✓	✕
			Significant life events	✓	✓	✓	✕	✕	✕	✕
			Social network nearby	✓	✓	✓	✕	✕	✕	✕
			Searching for social interaction	✕	✓	✓	✕	✕	✕	✕
			Volunteering	✓	✓	✓	✕	✕	✕	✕
			Shopping for family, friends, and neighbours	✓	✓	✓	✕	✕	✕	✕
Motivation	Reflective	Beliefs about capabilities	Lack of control	✓	✓	✓	✓	✓	✓	✓
			Self-efficacy	✓	✓	✓	✓	✓	✓	✓
			Self-confidence	✓	✓	✓	✓	✓	✓	✓
		Optimism	Family’s circumstances	✓	✓	✓	✓	✓	✓	✓
			Everyone needs to work together	✓	✓	✓	✓	✓	✓	✓
			A shared goal	✓	✓	✓	✓	✓	✓	✓
		Beliefs about consequences	Health consequences	✓	✓	✓	✓	✓	✓	✓
			Uncertainty of the health implications	✓	✓	✓	✓	✓	✓	✓
			Perceptions changed over time	✓	✓	✓	✕	✕	✕	✕
	Automatic	Reinforcement	Legally enforced	✓	✓	✓	✓	✓	✓	✓
			Parental discipline	✓	✓	✓	✓	✓	✓	✓

* NPI evaluated: 1 = stay at home, 2 = meeting others, 3 = physical distance, 4 = handwashing, 5 = avoid touching the face, 6 = covering a cough, 7 = cleaning the home. ** Indirect link to NPI 7, describes workplaces cleaning surfaces in the office (includes providing facilities) rather than the home.

## Data Availability

Data are available upon request to the corresponding author. However, we are unable to provide transcript material to ensure participant anonymity.

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
