# Peer review of "A Qualitative Study Evaluating the Factors Affecting Families’ Adherence to the First COVID-19 Lockdown in England Using the COM-B Model and TDF"

_ijerph, 2022, doi:10.3390/ijerph19127305_

Round 1

Reviewer 1 Report

The biggest concern I have is about the qualitative data analysis methodology in this study. The authors claimed that they applied the inductive method. However, the overall data analysis and literature review showed a deductive method. It looks like that the authors have a theoretical framework in their minds and then collected data to test it. I would suggest that the authors revisit their data analysis method and study design to better fit their aims.  

Author Response

In response to reviewer 1

Point 1: The biggest concern I have is about the qualitative data analysis methodology in this study. The authors claimed that they applied the inductive method. However, the overall data analysis and literature review showed a deductive method. It looks like that the authors have a theoretical framework in their minds and then collected data to test it. I would suggest that the authors revisit their data analysis method and study design to better fit their aims.  

Response to point 1: We feel our initial description was possibly confusing, and have revised it. At the time of our study there was limited understanding about a family’s adherence to COVID-19 guidance. Therefore, an inductive approach was appropriate. We used open ended questions, and produced themes that appeared salient in the data, without reference to the COM-B model. However, using a framework such as COM-B to subsequently help categorise the themes that we produced was helpful in allowing us to compare our results to the wider literature, and we therefore mapped them onto this framework. We have now removed the probably confusing references to inductive and deductive analysis and added that “we asked open ended questions about the topic area, which did not follow the COM-B and TDF theoretical framework” (line 139) to the interview outline section, in order to make clear our methods.

We also want to thank you for reviewing our manuscript and providing comments about our research. The suggestions we received were beneficial to the manuscript, and we hope the revised manuscript meets the criteria for publication.

Reviewer 2 Report

Overview:  Good overall study of some of the factors that influenced decisions of families during the Covid pandemic. 

Abstract:  Abstract is very good in its explanation and concise description of the study and findings.

Introduction:  Line 76: Please define the term “keyworker” with regards to “keyworker children”.  The term appears again in lines 79-80 and should be defined somewhere for those unfamiliar with the term.

The end of the introduction/literature review includes the theoretical approach to the study which is helpful, but there is no clear indication of the primary goal of the study or the research question that is being asked.  Please include a sentence within the last paragraph that clearly states the direction of this study and specific purpose. 

Materials and Methods:  Good overview of research methods

Results: The formatting of the end of the first paragraph should be revised.  The term “Table 2” appears as bold in line 193 and should be part of the previous sentence. 

Discussion:  Good overall discussion of findings, limitations and implications to the field, especially in England where the study was conducted. 

Author Response

Point 1: Introduction: Line 76: Please define the term “keyworker” with regards to “keyworker children”. The term appears again in lines 79-80 and should be defined somewhere for those unfamiliar with the term.

In response to point 1: We had defined “keyworker” in line 54. However, we have clarified this sentence (“such as children of keyworkers (children whose parents were working in roles critical to the COVID-19 response”) and other references to “keyworker children” (including lines 79-80) to ensure this term can be understood throughout the manuscript.

Point 2: The end of the introduction/literature review includes the theoretical approach to the study which is helpful, but there is no clear indication of the primary goal of the study or the research question that is being asked.  Please include a sentence within the last paragraph that clearly states the direction of this study and specific purpose. 

In response to point 2: We had a final statement that described the aims of the study. However, we had placed this after the diagram, we have moved this statement to follow on from the paragraph that describes the theoretical approach as we feel this is clearer.

Furthermore, we have added the below sentences to provide information about the specific purpose of our study.

“Understanding a family’s ability to change behaviour according to different NPIs is critical, particularly during school closures. Authorities need to incorporate this into their NPI strategy when controlling for COVID-19 and future disease outbreaks.”

Point 3: Results: The formatting of the end of the first paragraph should be revised.  The term “Table 2” appears as bold in line 193 and should be part of the previous sentence. 

In response to point 3: Thank you for highlighting the formatting error, we have corrected this in the manuscript.

We also want to thank you for reviewing our manuscript and providing positive comments about our research. The suggestions we received were beneficial to the manuscript, and we hope the revised manuscript meets the criteria for publication.

Round 2

Reviewer 1 Report

I have no further conerns.